# Asymptomatic neonatal herpes simplex virus infection in mice leads to persistent CNS infection and long-term cognitive impairment

**Abigail J. Dutton**[1,2‡], **Evelyn M. Turnbaugh**[1,2‡], **Chaya D. Patel**[1,2], **Callaghan R. Garland**[1], **Sean A. Taylor**[1], **Roberto Alers-Velazquez**[1], **David M. Knipe**[3], **Katherine M. Nautiyal**[4], **David A. Leib** [1]*

1 Department of Microbiology and Immunology, Geisel School of Medicine at Dartmouth, Lebanon, New Hampshire, United States of America, 2 Guarini School of Graduate and Advanced Studies at Dartmouth, Hanover, New Hampshire, United States of America, 3 Department of Microbiology, Blavatnik Institute, Harvard Medical School, Boston, Massachusetts, United States of America, 4 Department of Psychological and Brain Sciences, Dartmouth College, Hanover, New Hampshire, United States of America

‡ These authors contributed equally to this study.
* David.A.Leib@dartmouth.edu

## Abstract

Neonatal herpes simplex virus (nHSV) is a devastating infection impacting approximately 14,000 newborns globally each year. nHSV infection is associated with high neurologic morbidity and mortality, making early intervention critical. Clinical outcomes of symptomatic nHSV infections are well-studied, but little is known about the frequency of, or outcomes following, subclinical or asymptomatic nHSV. Given the ubiquitous nature of HSV infection and frequency of asymptomatic shedding in adults, subclinical infections are underreported and could contribute to long-term neurological damage. To assess potential neurological morbidity associated with subclinical nHSV infection, we developed a low-dose (100 PFU) intranasal HSV infection model in neonatal wild-type C57BL/6 mice. At this dose, HSV DNA was detected in the brain by quantitative PCR (qPCR) but was not associated with acute clinical signs of infection. However, months after neonatal inoculation with this low dose of HSV, we observed impaired mouse performance on a range of cognitive and memory tests. Memory impairment was induced by infection with either HSV-1 or HSV-2 wild-type viruses, indicating that the cognitive impairment associated with neonatal infection was not strain-specific. Maternal immunization reduced neonate central nervous system (CNS) viral burden and prevented offspring from developing neurological sequelae following nHSV infection. Altogether, these results support the idea that subclinical neonatal infections may lead to cognitive decline in adulthood and that maternal vaccination is an effective strategy for reducing neurological sequelae in infected offspring. These findings may have profound implications for understanding and modeling the etiology of human neurodegenerative disorders such as Alzheimer's Disease.

**Data availability statement:** All relevant data are within the manuscript and its Supporting Information files.

**Funding:** This study was supported by the National Institutes of Health (R01 EY09083 to DAL, P01 AI098681 to DAL, T32 AI007519 to AJD and CDP, and T32 AI007363 to EMT), The Munck-Pfefferkorn Education and Research Fund (5018 to DAL), The Rosaline Borison Memorial Fund Fellowship (to AJD), The Burroughs Wellcome Fund Postdoctoral Diversity Enrichment Program (PDEP) (to RAV), and by the Dartmouth International Vaccine Initiative (to DAL). The funders had no role in study design, data collection and analysis, decision to publish, or preparation of the manuscript.

**Competing interests:** The authors have declared that no competing interests exist.

## Author summary

Herpes simplex virus (HSV) establishes lifelong infection in the nervous system and is implicated in the pathogenesis of neurodegeneration. It is unknown, however, how the timing and nature of HSV infection impacts cognition in later life. Our work links neonatal HSV (nHSV) acquisition to later-life outcomes by showing that even subclinical neonatal infections can lead to cognitive impairment in adult mice. Moreover, we show that neurological sequelae following nHSV infection may be alleviated through maternal vaccination. Given high rates of asymptomatic HSV transmission, low-level nHSV exposures are likely occurrences and pose significant health risk. This work, therefore, introduces a model that demonstrates a possible connection between nHSV and neurodegenerative disorders such as Alzheimer's Disease.

## Introduction

Herpes simplex virus (HSV) is a neurotropic DNA virus that establishes latency within sensory neurons [1–3]. Periodic reactivation of HSV may be asymptomatic, result in recurrent vesicular lesions, or lead to more severe infections including encephalitis and stromal keratitis [4]. These diverse disease manifestations usually follow anterograde transport of the virus within neuronal axons from sites of latency to sites of recrudescence [5]. Some of the most severe outcomes follow primary neonatal infections in which direct damage and associated pathological inflammation can lead to severe and permanent neurological impairment and death [6,7]. Antiviral therapy decreases the frequency and severity of both symptomatic reactivation and neonatal disease, but there is no cure or vaccine for HSV infection [8,9]. Current estimates suggest that two thirds of the global population under the age of 50 are infected with HSV, underscoring the large burden of both disease and potential asymptomatic transmission [10].

Consistent with life-long persistence in sensory neurons and its ability to enter and cause inflammation within the brain, HSV has been implicated in a number of neurologic diseases, including Alzheimer's disease (AD) [11–15]. AD is a chronic neurodegenerative disorder and the most common form of dementia in elderly populations [16]. AD and related dementias affect over 40 million people globally and are characterized by memory loss, anxiety, agitation, and depression [16,17]. These symptoms manifest from chronic inflammation, accumulation of neurotoxic protein, and neuronal loss in the central nervous system (CNS) [18]. The chronic CNS inflammation induced by certain neurotropic pathogens supports the hypothesis that viral infection may play a role in the development of AD [19]. Post-mortem brains exhibiting a significant level of AD-associated pathology were found to have a high prevalence of HSV DNA associated with accumulated neurotoxic protein, supporting a causal link [20]. The presence of HSV DNA in the CNS of individuals with AD, along with the propensity of HSV to cause neurologic morbidity, have implicated HSV in the pathogenesis of AD but possible molecular mechanisms have yet to be elucidated [12,13].

Mouse models have previously been utilized to investigate the possible connection between HSV and AD [21,22]. HSV-1-infected mice show CNS damage, inflammation, and an AD-like phenotype after multiple viral reactivations [21,22]. In this model, a high dose of HSV followed by multiple rounds of heat-induced reactivation results in neurologic impairment. It is likely that many human HSV infections result from lower infectious doses during both symptomatic and asymptomatic shedding and transmission. Asymptomatic shedding, which

occurs during both acute infection and viral reactivation events, contributes to high rates of infection and exposure, and makes incidence of infection in the human population difficult to track [23–26]. The majority of individuals who are seropositive for HSV are unaware of their infection [27,28]. We therefore developed a mouse model to study asymptomatic, subclinical HSV infection and ensuing morbidities.

To investigate the longitudinal implications of HSV infection, we utilized a neonatal mouse model. During the neonatal period, host immune and nervous systems are still developing, rendering neonates particularly vulnerable to infection [29–32]. Early life exposure to infectious or toxic insults can perturb development and lead to neurologic and psychiatric disease in later life [32,33]. While serious acute cases of neonatal HSV (nHSV) are well studied, the life-long burden of asymptomatic nHSV infections remains unknown. The incidence of asymptomatic nHSV has not been determined due to low rates of maternal testing [34], and because nHSV is often not diagnosed unless symptoms present [35–37]. Given the high prevalence of HSV infection in the adult population, asymptomatic HSV infection could be an unrecognized source of morbidity that may have lifelong implications for infected neonates [38,39].

In mice, low-dose nHSV infection causes long-lasting, anxiety-like behavior during adolescence [40,41]. In this study, we aimed to characterize an asymptomatic nHSV infection model and investigate the role of neonatal viral infection in cognitive decline during adulthood. We observed that low-dose nHSV infection of wild-type mice led to long-term memory-loss and decreased behavioral flexibility. The behavioral sequelae induced by nHSV were prevented if mice were born to HSV-immunized dams, supporting the idea that the neurological changes were due to infection. Importantly, in this era of vaccine hesitancy [42,43], we showed that vaccine challenge alone did not cause cognitive decline. Together, findings from this work support a link between cognitive decline and early life exposure to pathogens, and further support the hypothesis of a connection between HSV infection and AD.

## Results

### Characterization of a low-dose neonatal HSV infection model

To model the impact of neonatal exposure to small doses of HSV, we tested escalating viral doses with a goal of achieving CNS infection without acute clinical signs (Fig 1A). On day 1 of life (P1), litters of mice were infected intranasally (i.n.) with HSV or mock-infected with cell lysate. At 5 days post-infection (dpi), we collected brains of inoculated mice to determine viral burden. Mice that were infected with 100 plaque-forming units (PFU) strain 17 (st17) HSV-1 showed normal weight gain (Fig 1B) and no clinical signs of acute infection. Notably, over 90% of 100 PFU nHSV-infected pups survived, while only 40% of pups survived infection with 1000 PFU HSV-1 (Fig 1C). To determine the possible extent of HSV spread in the CNS following i.n. infection, we sliced brains of pups infected with 1000 PFU HSV-1 and assessed viral antigens by immunofluorescence at 5 dpi (Fig 1D). Specifically, we stained for the presence of HSV-1 glycoprotein C (gC), a viral envelope protein, to quantify productive viral infection in the CNS at dpi 5. Measurable fluorescence, indicating presence of HSV antigen, was detected in the brainstem, cerebellum, midbrain, diencephalon, isocortex, and olfactory bulbs of a subset of 1000 PFU nHSV-infected pups (Fig 1E). Although not every infected pup had detectable antigen in their brain, these data demonstrated that i.n. infection with 1000 PFU HSV-1 resulted in broad CNS infection by 5 dpi.

We used qPCR to evaluate CNS infection following 100 PFU infection of pups. HSV genomes were detectable in the midbrain of a subset of infected pups at 5 dpi (Fig 1F). As expected, higher viral inocula resulted in greater average genome loads. HSV genomes were

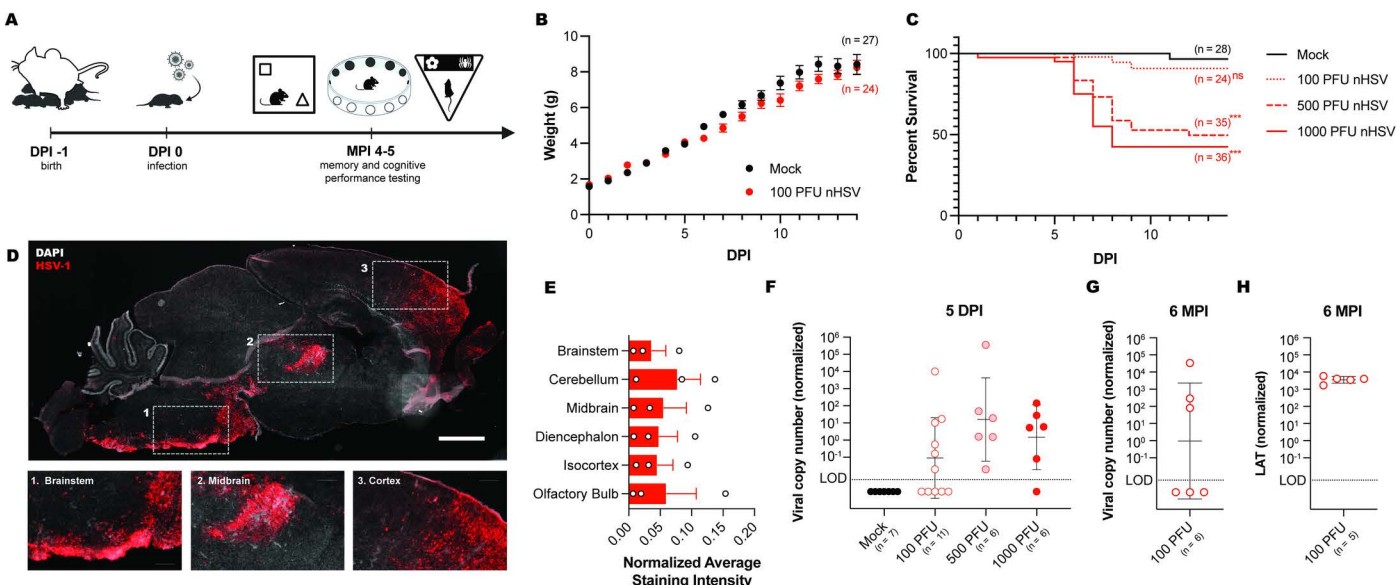

**Fig 1. Characterization of the low-dose neonatal HSV infection model. (A)** Schematic of the timeline of infection and behavioral assessment. One-day-old C57BL/6 mice were intranasally (i.n.) infected with escalating doses of strain 17 (st17) HSV-1 or virus-free lysate "mockulum". At 4-5 mpi, nHSV and mockulum-treated mice were assayed for memory and cognitive performance deficits using the novel object recognition (NOR) task, the novel object location (NOL) task, the modified Barnes maze (MBM), and the different paired-associate learning (dPAL) task. **(B)** Weights of pups i.n. infected with 100 PFU HSV-1 (24) or mock-infected (27). **(C)** Survival of neonatal mice i.n. infected with mockulum, 100 PFU, 500 PFU, or 1000 PFU HSV-1. **(D)** Representative brain slice of a 5 dpi 1000 PFU HSV-1 infected mouse stained with DAPI (white) and antibody against HSV glycoprotein C (gC; red). Scale bar = 1 mm. **(E)** Normalized average staining intensity of gC staining in brains of 5 dpi 1000 PFU HSV-1 infected mice. **(F)** HSV genome copy number in the midbrain of 5 dpi mock-, 100 PFU-, 500 PFU-, and 1000 PFU-infected mice, relative to mouse single-copy adipsin gene. **(G)** HSV genome copy number in the midbrain of 6 mpi 100 PFU HSV-1-infected mice, relative to mouse single-copy adipsin gene. **(H)** LAT quantification in the midbrain of 6 mpi 100 PFU HSV-1-infected mice, relative to mouse single-copy adipsin gene. Statistical significance was determined by comparing infected groups to mock-infected animals using two-way ANOVA (B), log-rank (Mantel-Cox) tests (C), and one-way ANOVA (E, F). ns = not significant, ***p < 0.001, ns = not significant. Error bars represent standard error (C, E; SEM) and standard deviation from the geometric mean (F,G). Fig graphics created in BioRender. Leib, D. (2025) https://BioRender.com/s18a614.

detected in the trigeminal ganglia and all regions of the brain assayed including the olfactory bulbs, cortex, and hindbrain (S1 Fig). We thereby determined that 100 PFU of HSV-1 was an effective dose to establish detectable nervous system infection at the neonatal timepoint without causing significant mortality or morbidity, thus establishing the parameters for studying the effects of subclinical nHSV infection on adult cognition. To assess viral persistence in the brains of nHSV-infected mice, we quantified viral DNA and HSV latency-associated transcripts (LATs) in the brains of 100 PFU-infected mice at 6 mpi. The gC-positive staining and viral DNA quantified at 5 dpi was evidence of late (L)-stage lytic infection, consistent with expression of the full kinetic range (immediate early, early, and late) of viral genes. We wished, therefore, to assess the adult CNS for the presence of HSV genomes and latency-associated transcripts (LATs) in order to confirm the presence of transcriptionally active latent HSV infection following neonatal inoculation. Notably, at 6 mpi, a subset (3/6) of animals had detectable viral genomes in the midbrain region (Fig 1G) and in a separate experiment, 5/5 midbrains were positive for latency-associated transcript (LAT, Fig 1H). Viral genomes and LATs were detected in all areas of the brain assayed in at least a subset of nHSV infected animals (S1E and S1F Fig). These data demonstrate that HSV persists in the mouse CNS in a transcriptionally active state for months after initial neonatal infection. Together, these data show that low-dose HSV infection can lead to lasting CNS infection in neonatally infected mice, warranting further assessment of the impact of nHSV infection on adult neurocognitive health.

## Low-dose nHSV infection causes novel recognition impairment in adulthood

Mice infected with nHSV-1 demonstrate anxiety-like behavior in the open field test (OFT) 5 weeks after infection [40,41]. Given the possible link between HSV infection and Alzheimer's disease [11–13], we tested whether neonatal infection with 100 PFU HSV-1 would also lead to memory loss in adult mice. To examine this, we used the novel object recognition (NOR) task to assess recognition memory at 4–5 months post infection (mpi). The NOR task is derived from the visual paired-comparison paradigm used in human studies to assess recognition memory in the context of disease and/or aging [44]. In the NOR task, on day 1 (familiarization phase), mice were allowed to explore two identical objects placed in an open enclosure for 10 minutes (Fig 2A). On day 2 (testing phase), one object was replaced by a novel object and interaction time with each object was recorded to assess recognition memory, measured by more time exploring the novel object [45]. At 4–5 mpi, mock-infected mice showed greater exploration of the novel object than of the familiar object, as expected. nHSV-infected mice, however, did not show increased exploration of the novel object, suggesting impaired memory recognition (Fig 2A lower panel). The average discrimination index (DI; a metric of relative preference [46,47]) for mock-infected mice was significantly higher than the average DI for nHSV-infected animals ($t(36) = 2.900$, $p = 0.0063$), indicating that nHSV infection led to impaired NOR performance at 5 months of age.

Given that HSV-2 predominates as the causative agent of nHSV in many parts of the world [10,48], we wished to examine whether nHSV-2 infection could cause learning and memory deficits as shown for HSV-1. We adjusted the i.n. dose of HSV-2 strain 333 for neonatal mice to 75 PFU because HSV-2 is generally more virulent than HSV-1 and a dose reduction was required to attain an asymptomatic infection [49,50]. Consistent with the HSV-1 results, there was a significant effect of infection on novel object discrimination in the NOR task ($F(2,47) = 4.263$, $p = 0.0199$; S2A Fig). The average discrimination index of nHSV-2 infected mice was significantly lower than the average discrimination index of mock-infected mice ($p = 0.0466$, Tukey's multiple comparisons test), but was not significantly different from the average discrimination index of nHSV-1 infected mice ($p = 0.9673$, Tukey's multiple comparisons test). This indicated that, consistent with 100 PFU st17 HSV-1 neonatal infection, neonatal HSV-2 infection was associated with memory impairment (S2A Fig). Having established that HSV-1 and HSV-2 cause similar behavioral impairment on the NOR task, we opted to use only HSV-1 moving forward in this study.

To test the effect of nHSV infection on spatial memory, we used the novel object location (NOL) task [46,51]. Spatial memory impairment is one of the hallmark deficits associated with AD, shared by over 60% of those with a diagnosis [52,53]. This assay was performed similarly to the NOR task, except an object was moved to a novel location during the testing phase rather than being replaced by a novel object (Fig 2B). Additionally, the time between familiarization and testing was reduced to 1 hour to assess short-term spatial recognition memory. Mock-infected mice spent more time interacting with the object in its novel location than with the object in its familiar location, demonstrating intact spatial memory (Fig 2B lower panel). In contrast, nHSV-infected mice spent more time interacting with the object in its familiar location than with the object in its novel location. Overall, nHSV-infected mice demonstrated impaired location recognition memory compared to mock-infected mice (Fig 2B lower panel; $t(43) = 2.090$, $p = 0.0426$). Interestingly, this NOL recognition impairment mirrors the performance of 7-month-old female 5xFAD mice on the NOL task at baseline (Fig 2B lower panel). 5xFAD mice express 5 familial AD mutations resulting in overexpression of APP and PSEN1 and exhibit early onset amyloid pathology and cognitive deficits [54,55]. Both nHSV-infected

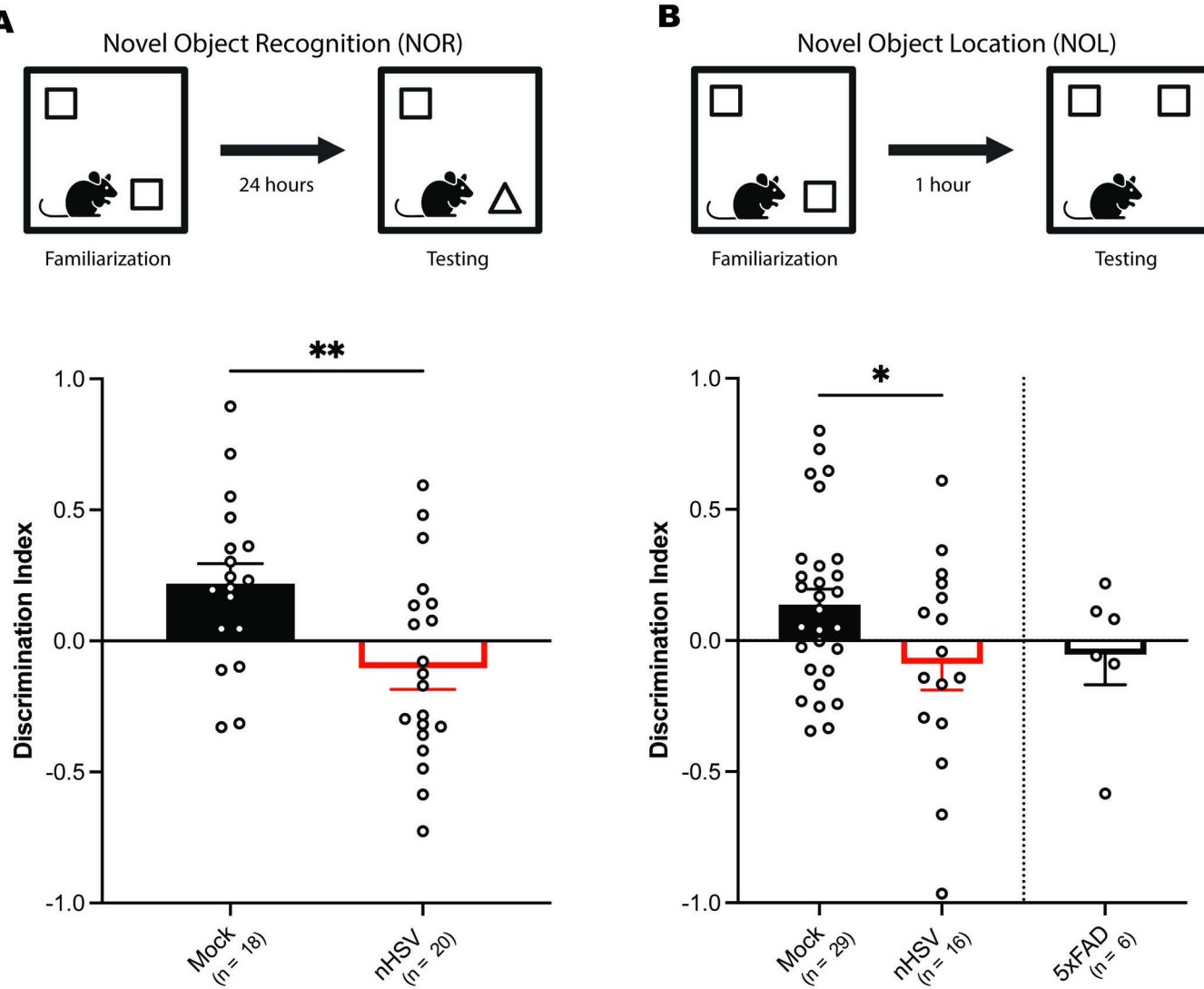

**Fig 2. Low-dose nHSV infection leads to impaired novelty recognition in adult mice. (A)** Schematic of novel object recognition (NOR) task (top), and performance of mock-infected (n = 18) and strain 17 (st17) nHSV-infected (n = 20) mice on the NOR task at 5 mpi (below). Bar graph shows the mean discrimination index for the novel object. **(B)** Schematic of novel object location (NOL) task (top) and performance of 5 mpi mockulum-treated mice (n = 29), 5 mpi st17 nHSV-infected mice (n = 16), and 7-month-old uninfected female 5xFAD mice (n = 6) on the NOL task (below). Bar graph shows the mean discrimination index for the object in a novel location. Discrimination index = (Time$_{novel}$ − Time$_{familiar}$)/Time$_{total}$. Statistical significance was determined by unpaired t-tests. *p < 0.05, **p < 0.01. Error bars represent standard error (SEM). Fig graphics created in BioRender. Leib, D. (2025) https://BioRender.com/k15h873.

mice and 7-month-old female 5xFAD mice did not show preference for the object in its novel location, indicating shared spatial memory impairment.

## Low-dose nHSV infection causes impairment in behavioral flexibility in adulthood

To further define the behavioral impairment associated with nHSV infection, we used the modified Barnes maze (MBM). The MBM is a spatial learning task in which mice use visual orientation cues to navigate a circular enclosure to find an escape hole (Fig 3A) [56,57]. A reversal phase allows for assessment of reversal learning and behavioral flexibility encoded by

the prefrontal cortex [58] (PFC). nHSV-infected mice performed similarly to mock-infected mice during the training phase of the assay, with all mice becoming faster at locating the exit hole over four days of training (Fig 3B; main effect of training day, $F_{(1.35,17.54)} = 8.67$, $p < 0.0001$; no effect of infection status, $F_{(1,13)} = 0.0107$, $p = 0.9192$). This demonstrated that there was no effect of nHSV infection on learning or navigation during the training phase of the MBM task.

During the reversal phase of the MBM, both nHSV-infected and mock-infected mice showed improvement in time to exit the maze across subsequent trials (Fig 3B; main effect of training day, $F_{(1.997,25.70)} = 5.35$, $p = 0.0116$). However, planned post-hoc analyses revealed that infection status impacted mouse performance in the maze during the first trial of reversal testing when the escape hole was moved to the opposite side of the arena. On the first trial of reversal testing, nHSV-infected mice took longer to escape than mock-infected mice ($t(13) = 2.590$, $p = 0.0224$). In addition, nHSV-infected mice traveled significantly more distance to locate the escape hole ($t(13) = 2.744$, $p = 0.0167$), and made significantly more errors per trial ($t(13) = 5.610$, $p < 0.0001$) than mock-infected mice (Fig 3E and 3F). The same was true for nHSV-2 infected mice (S2B–S2D Fig). Finally, during the first reversal testing session, nHSV-infected mice committed more errors in the quadrant containing the original escape hole than mock-infected mice ($t(13) = 5.610$, $p < 0.0001$; Fig 3G). This pattern was suggestive

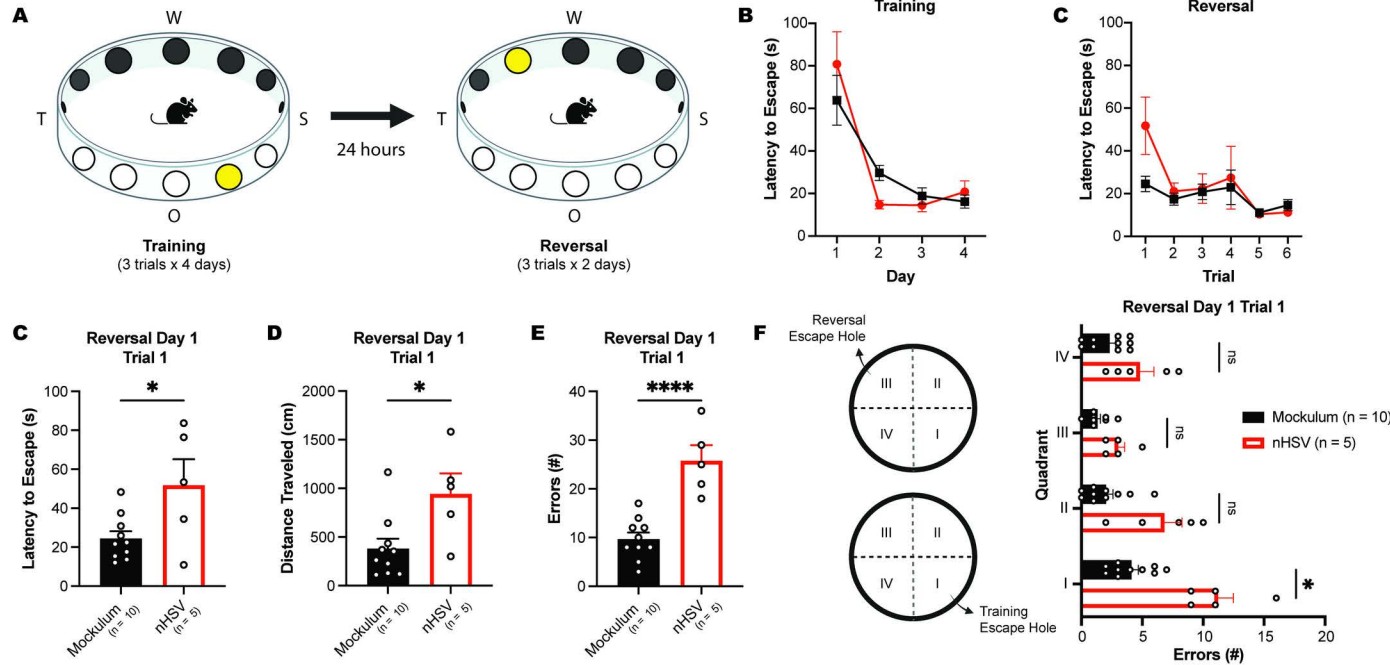

**Fig 3. Low-dose nHSV infection leads to impaired behavioral flexibility in adult mice. (A)** Schematic of modified Barnes maze (MBM) test. Over the course of 4 days of 3 trials per day, mock-infected and 100 PFU HSV-1 strain 17-infected mice were trained in a MBM arena to locate, and exit through, an escape hole using visual cues ("W", "S", "O", and "T") for orientation. Following training, the location of the exit hole was reversed, and mice were tested for their ability to locate the reversal phase exit. **(B)** Mean latency to escape per day during the training phase of the MBM. **(C)** Mean latency to escape per trial during the reversal testing phase of the MBM. **(D)** Mean latency to escape on the first trial of reversal testing in the MBM. **(E)** Mean distance traveled in centimeters (cm) during the first trial of reversal testing in the MBM. **(F)** Mean errors per trial committed during the first trial of reversal testing in the MBM. **(G)** Quadrant designations (left) for the location of the escape hole during training (I) and reversal (III) phases of the MBM. Mean number of errors per quadrant (right) committed during the first reversal trial on day one of reversal testing. Statistical significance was determined by two-way ANOVA with a repeated measures design (B, C) and unpaired t-tests (D-G). ns = not significant, *p < 0.05, ***p < 0.001, ****p < 0.0001. Error bars represent standard error (SEM). Created in BioRender. Leib, D. (2025) https://BioRender.com/c44t589 (A) and https://BioRender.com/d82i762 (F).

of increased perseveration and impaired behavioral flexibility, which is consistent with nHSV-induced PFC impairment [59]. Importantly, nHSV-infected mice performed similarly to mock-infected mice on all subsequent reversal trials, indicating that the ability to re-learn the new location of the exit hole was not impacted.

## Low-dose nHSV infection causes impairment in hippocampal-dependent spatial memory in adulthood

To probe hippocampal-dependent behavior, we tested mice using the different Paired-Associate Learning (dPAL) task on a touch screen behavioral operant system [60]. The dPAL task assesses the mouse's ability to indicate the correct location of a trained visual stimulus, thereby testing hippocampal spatial memory. A similar version of this task is used to assess memory cognition in AD patients [61]. In mice, dorsal hippocampal lesions impair performance on this task but do not disrupt task acquisition [60]. Following operant training, we tested mock-infected and nHSV-infected mice at 5 mpi. During each testing session, mice completed up to 36 trials in which a correct response was noted as an interaction with the shape (spider, flower, or plane) in its correct location (Fig 4A). Both groups exhibited evidence of learning over time and reached an average of approximately 75% correct responses by the final day of testing (day 40). The

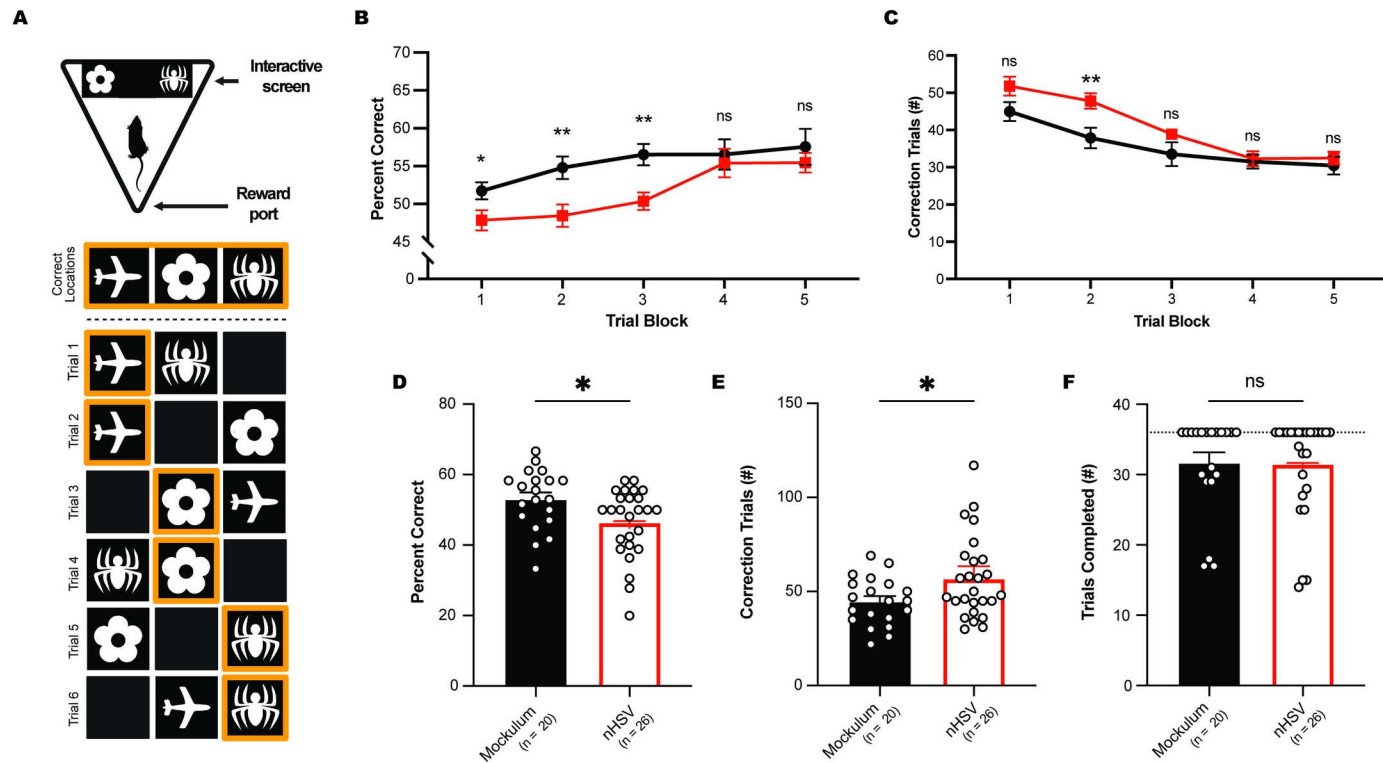

**Fig 4. Low-dose nHSV infection leads to impaired associative memory in adult mice.** (A) Schematic of a mouse completing the different paired-associate learning (dPAL) paradigm in a Bussey box chamber (top), and a diagram representing all possible combinations of test stimuli and correct responses (orange; bottom). (B) Average performance (% correct on first attempt) of mock-infected (20) and strain 17 (st17) nHSV-infected (26) mice during the first 5 trial blocks. 1 block = 3 testing sessions = 108 trials completed across 3 days. (C) Average number of correction trials per testing block during the first 5 blocks. (D) Average performance (% correct on first attempt) on day 1 of dPAL testing. (E) Average number of correction trials on day 1 of dPAL testing. (F) Average number of total trials completed on day 1 of dPAL testing. Statistical significance was determined by two-way ANOVA with repeated measures design and post-hoc Sidak multiple comparisons tests (B, C) and unpaired t-tests (D-F). ns = not significant, *p < 0.05. Error bars represent standard error (SEM). Fig graphics created in BioRender. Leib, D. (2025) https://BioRender.com/v77i905 (A, top), https://BioRender.com/t51l472 (A, bottom).

nHSV-infected group, however, showed a reduced proportion of correct responses compared to the mock-infected group during initial days of testing. There was a significant effect of training (F(4,175) = 9.300, $p < 0.0001$) and of infection status (F(1,44) = 7.299, $p = 0.0098$) on the percentage of correct responses given by mice in the first five testing blocks (Fig 4B). In addition, nHSV-infected mice completed a greater number of correction trials than mock-infected mice, indicating an increased error rate (Fig 4C). There was a significant effect of training (F(3.426, 150.7) = 25.5, $p < 0.0001$) and infection status (F(1,44) = 6.956, $p = 0.0115$) on the number of correction trials made during the first five blocks of testing. Together, these effects indicated that both groups of mice improved in dPAL performance over time. That said, nHSV-infected mice had a lower performance (% correct) and completed a greater number of correction trials than mock-infected mice during early testing timepoints. To further explore the initial deficit in nHSV-infected mice, we analyzed the performance of mice following their first exposure to the dPAL paradigm. On the first day of dPAL testing, nHSV-infected mice achieved a significantly lower percentage of trials correct than mock-infected mice (Fig 4E; $t(44) = 2.376$; $p = 0.0219$) and completed significantly more correction trials (Fig 4F; $t(44) = 2.202$, $p = 0.0329$). Importantly, both groups completed approximately the same number of trials during the first testing session indicating that performance deficits were not due to differences in exposure to the paradigm (Fig 4F; $t(44) = 0.0722$, $p = 0.9428$). By the fifth block of testing, nHSV-infected mice performed similarly to mock-infected mice, indicating that despite initial deficits, nHSV-infected mice learn the task and improve in performance over time (Fig 4B and 4C). Taken together, mouse performance on the NOR, NOL, MBM, and dPAL tasks demonstrated that neonatal infection with HSV leads to changes in memory and cognitive behavior analogous to those seen in AD [62].

## HSV-specific maternal immunity protects against nHSV-1 induced behavioral morbidity

Maternal immunity is critical to the outcome of nHSV morbidity and mortality in mice [41,63]. Both passive and active immunization confer HSV-specific antibody protection in dams that lowers viral titers and anxiety behavior in adult offspring that were neonatally challenged with HSV [41,63]. Given this, we determined the ability of maternal vaccination to provide lasting protection against CNS infection and behavioral morbidity associated with nHSV infection (Fig 5A). First, dams were vaccinated with live, replication-defective vaccine strain HSV-2 (*dl*5-29) [64,65] via intramuscular injection prior to parturition. To investigate whether maternal vaccination could protect against nHSV CNS infection, we collected brain slices from 5 dpi 1000 PFU nHSV-infected offspring of *dl*5-29 vaccinated dams. In contrast to brains of pups from unvaccinated dams (Fig 1C), brains of pups from vaccinated dams showed no detectable viral antigen in the CNS at 5 days post-nHSV infection (Fig 5B). ImageJ analysis revealed a normalized average staining intensity that was statistically indistinguishable from background. In addition, HSV DNA was undetectable by qPCR in brains of 100 PFU HSV-infected pups from vaccinated dams (Figs 5C and S3). These results showed that maternal vaccination with *dl*5-29 was sufficient to prevent detectable acute CNS infection in offspring at 5 dpi following intranasal inoculation with 100 PFU HSV-1. Whether virus failed to enter the CNS during initial infection or was cleared before establishment of latency in the brain is unknown.

At four-to-five months post-infection, nHSV-infected offspring of vaccinated dams were tested using the NOR task to determine whether maternal vaccination could also protect offspring from the neurological morbidity observed in our model. Neonatally HSV-challenged offspring of seronegative, unvaccinated dams spent less time with the novel object than with the familiar object, while offspring of seropositive, vaccinated dams exhibited intact object recognition memory behavior (Fig 5D). There was a significant effect of infection

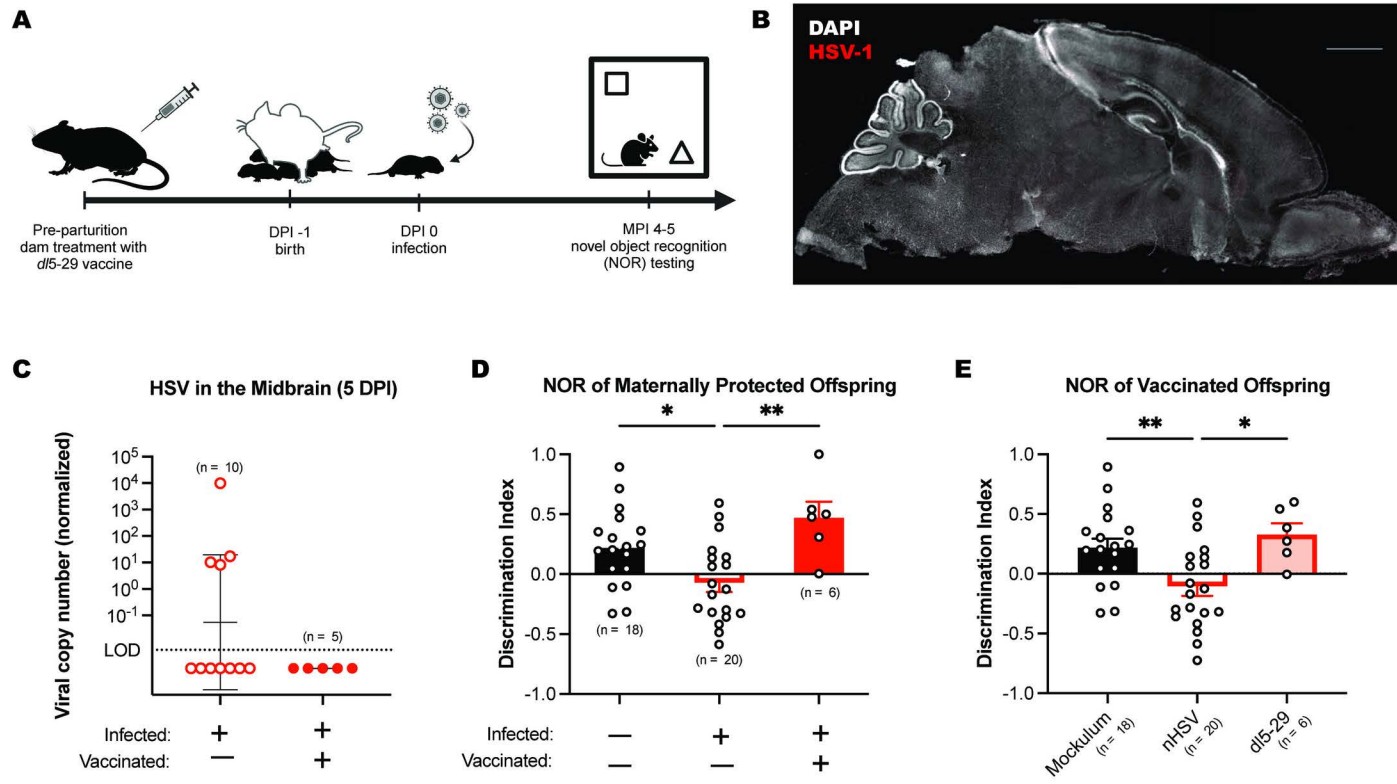

**Fig 5. Maternal vaccination protects against HSV CNS infection and nHSV-induced behavioral morbidity. (A)** Schematic showing the timeline of dam treatment, neonatal infection, and behavioral assessment using the NOR task. **(B)** Representative brain slice of a 5 dpi 1000 PFU nHSV-1 infected mouse pup from a *dl*5-29 vaccinated dam stained with DAPI (white) and antibody against HSV glycoprotein C (gC; red). Scale bar = 1 mm. **(C)** HSV genome copy number in the midbrain of 5 dpi 100 PFU nHSV-1 infected mice from naïve and *dl*5-29 vaccinated dams, relative to mouse single-copy adipsin gene. **(D)** Average discrimination index of mock-infected and strain 17 (st17) nHSV-infected offspring of naïve or *dl*5-29 vaccinated dams at 5 mpi on the NOR task. **(E)** Average discrimination index of mice neonatally infected with replication-deficient HSV vaccine strain *dl*5-29 at 5 mpi on the NOR task compared to mice neonatally infected with 100 PFU HSV-1 or mockulum (Vero cell lysate). Data are shown as average discrimination index for the novel object during the testing phase of the NOR task. Discrimination index = (Time$_{novel}$ − Time$_{familiar}$)/Time$_{total}$. Statistical significance determined by unpaired t-test (C) or one-way ANOVA with Dunnett's multiple comparisons post-hoc tests (D-E). ns = not significant, *p < 0.05, **p < 0.01. Error bars represent standard deviation from the geometric mean (C) and standard error (D, E; SEM). Fig graphics created in BioRender. Leib, D. (2025) https://BioRender.com/m43o918 (A).

and vaccination status on NOR performance (F(2,40) = 7.422, p = 0.0018). nHSV-infected offspring of unvaccinated dams demonstrated impaired novel object recognition relative to nHSV-infected offspring of vaccinated dams (p = 0.0022, Dunnett's test for multiple comparisons). Therefore, the data show a protective effect of maternal vaccination on novel object recognition in nHSV-infected offspring.

Finally, to assess the risks of the live replication-defective vaccine virus itself conferring memory impairment, we i.n. infected neonatal mice with 100 PFU of the replication-defective HSV vaccine strain *dl*5-29 and followed up with NOR testing. At 4 mpi, vaccination status had a significant effect on novel object discrimination (F(2,41) = 6.382, p = 0.0039). Specifically, adult mice that were neonatally challenged with *dl*5-29 demonstrated a significantly higher average discrimination index than nHSV-1 infected mice (p = 0.0198, Dunnett's multiple comparisons test), demonstrating intact novel object recognition (Fig 5E). The discrimination index of *dl*5-29-treated mice was statistically similar to that of mock-infected mice (p = 0.7571). This lends support for its safety as a vaccine, and together these data indicated that maternal immunity is a critical determinant of protection against HSV-induced behavioral impairment in offspring.

## Discussion

Herpes viruses are a highly prevalent part of the human virome. HSV CNS infections are common causes of encephalitis and neurologic dysfunction [66–69], but the contribution of HSV infections to long-term memory and cognitive decline remains underexplored. In this study, we developed a mouse model of asymptomatic neonatal infection and demonstrated that nHSV-1 and nHSV-2 cause behavioral impairment months after initial infection. Importantly, we showed that these morbidities occur in the absence of acute clinical signs, such that even subclinical infection may lead to subsequent cognitive impairment. At 6 mpi, we showed that a subset of nHSV-infected mice had detectable viral genome and latency-associated transcripts (LATs) in the CNS, suggesting that behavioral impairment may correlate with persistent infection of the nervous system. We further demonstrate that in our model, maternal vaccination protects against CNS infection and behavioral morbidity following nHSV infection. These studies provide evidence for a link between viral infection and neurologic dysfunction and introduce a novel model for the study of viral-driven neurodegeneration.

Our results are consistent with previous studies that showed cognitive decline, hippocampal neurodegeneration, and neuroinflammation in mice with recurrent HSV-1 infections [13,21,22]. Further studies have also shown behavioral impairment associated with SARS-CoV-2 [70], West Nile Virus (WNV) [71,72], and Zika virus infection [72,73]. Prenatal and neonatal viral infections have specifically been linked to behavioral impairment [70,73], suggesting that viral infection during developmental periods may warrant special consideration. The exquisite sensitivity of neonatal mice to infection likely predisposes the CNS to local inflammation and cognitive impairment that is clinically inapparent until later life. Our hypothesis is that even small perturbations during the neonatal period can affect neurodevelopment at a period in which the brain and immune system are rapidly changing. Many known maternal and fetal insults are accompanied by long-term behavioral alterations in offspring [29,32,70,73]. Prenatal alcohol exposure, for example, causes changes in neuronal migration in the prefrontal cortex leading to behavioral deficits [74]. It is possible that HSV invasion of the CNS during development, as well as the ensuing immune response, can shift neuronal differentiation or migration, leading to lifelong changes in the brain. If this neonatal model is translatable to human infections, the public health implications are profound.

Our model stands out from previous studies in two important ways. First, we detected behavioral changes without induction of reactivation following thermal stress, as other HSV studies have used [22]. This suggests that induced reactivation may not be necessary for cognitive decline, although it is possible that spontaneous reactivation occurs during our 6 month experimental window. Second, we infected mice with low doses of HSV during the neonatal period, in contrast to studies that infect with high doses of virus during adulthood. A low-dose, neonatal infection model allowed us to examine the lifelong, accumulated effect of subclinical, asymptomatic infections on cognition.

Our previous work has shown that nHSV induces anxiety-like behavior [40,41]. The work described here characterizes cognitive impairment associated with nHSV using NOR, NOL, the MBM, and dPAL tests. On the NOR and NOL tests, infected mice showed impaired object and spatial novelty recognition, indicating broad memory deficit. On the MBM, infected mice showed impaired performance on the reversal phase of testing, indicating a deficit in PFC-mediated behavioral flexibility [59]. On dPAL, infected mice demonstrated impaired hippocampal-associated learning and memory, relative to mock-infected mice. Together, performance on these behavioral tasks points to areas of the brain that may be particularly vulnerable to neonatal HSV infection or consequential inflammation and neurodegenerative processes such as those seen in AD. Impaired neurogenesis, neuronal loss, and accumulation of misfolded protein are additional hallmarks of AD which we have yet to investigate.

Ongoing work will characterize the lifelong neuroanatomical consequences of neonatal HSV exposure as well as the mechanism of neurodegeneration following viral infection.

The precise role of persistent virus in the development of cognitive decline in nHSV-infected mice remains an open question. At 6 mpi, we detected viral genome and LATs in the brains of a subset of animals that had been neonatally infected with HSV. This finding suggests that transcriptionally active genomes, reactivation, or both could be associated with behavioral impairment. Future investigations should aim to clarify the roles latent and replicating HSV play in the development of nHSV-associated neuroimpairment.

Our lab previously showed that maternal HSV-specific antibody is transferred to the nervous system of the fetus in humans and mice [63], and in humans, maternal antibodies protect against neonatal infection [75]. Additionally, we demonstrated that maternal-derived antibodies prevent nHSV-induced behavioral morbidity [41]. In the current study, maternal vaccination was sufficient to protect against nHSV-induced cognitive impairment in mice. Maternal immunity, therefore, is crucial to protect against nHSV infection and its related sequelae, and there is ongoing work to develop therapeutic and prophylactic antibodies and vaccines to protect against nHSV [76]. If nHSV infections lead to behavioral impairment, the safety of live attenuated vaccine strains must be carefully considered. Importantly, the HSV-2 *dl*5-29 replication-deficient vaccine strain itself did not lead to memory impairment after neonatal infection, supporting the idea that this represents a safe vaccine candidate [77]. Beyond vaccines, many companies are developing HSV-based oncolytic and gene delivery vectors for EB, melanoma, and other diseases [78–81]. It follows from this study that these vectors should be analyzed for their potential to enter the CNS and cause behavioral impairment.

Over 60% of the population is seropositive for HSV-1, so it is important to consider both therapeutic and prophylactic approaches for addressing HSV-induced cognitive impairment [48]. The HSV antiviral valacyclovir is currently in Phase 2 clinical trials for treatment of AD by the National Institute on Aging (NCT03282916). In the double-blind trial, 130 patients with mild AD will be given valacyclovir or placebo daily for 78 weeks and monitored for cognitive function and accumulation of amyloid and tau protein. Results from this trial have the potential to inform our understanding of the relationship between viral Infection and AD, as well as provide a model for testing novel therapeutic approaches for the treatment of AD. A direct link between HSV and neurodegeneration has remained elusive since the early 1990s when the association was first proposed [12]. The observation that HSV-specific immunity protects against cognitive decline following nHSV infection strengthens the argument that HSV may play a role in neurodegeneration.

Previous research linking HSV with AD points to three hypotheses for an infectious etiology of AD. First, multiple reactivations of latent virus could produce chronic brain disease and progressive injury [22]. Second, acute infection creates a pro-inflammatory response that exacerbates chronic neurodegeneration [82]. Third, viral infection seeds Aβ in the brain and viral antagonism of autophagy may contribute to its accumulation [83]. Our current work cannot exclude any of these hypotheses. Our observation that HSV genomes and transcripts are detectable in the CNS of low-dose nHSV-infected mice at 6 mpi strongly supports a link between viral neuroinvasion and behavioral impairment. However, in the absence of data showing direct evidence of reactivation or inflammation associated with viral genome in the CNS, the exact mechanism of impairment remains unknown. This neonatal model provides a relatively straightforward framework with which to define mechanisms by which HSV infections lead to cognitive decline.

To our knowledge, our work is the first to show memory impairment following low-dose neonatal infection. Given that 70% of the HSV-1 seropositive population sheds asymptomatically at least once per month [25], there is a significant possibility of low-level neonatal exposure. Therefore, our work highlights a window of vulnerability to HSV infection that has not been investigated in previous studies of viral-induced neurodegeneration. Although the

host response following low-dose neonatal infection remains to be fully explored, our work elucidates the potential for subclinical, asymptomatic infections to cause long-term changes to behavior and cognition.

## Materials and methods

### Ethics statement

All animal experiments were conducted with approval from the Institutional Animal Care and Use Committee (IACUC) at the Geisel School of Medicine at Dartmouth (IACUC protocol number 00002151 m1, expiration date 05/17/2027). This institution has an Animal Welfare Assurance on file with the Office of Laboratory Animal Welfare, assurance number D16-00166 (A3259-01). Animal experiments were performed in an Association for Assessment and Accreditation of Laboratory Animal Care (AALAC)-approved facility by certified staff, following the basic principles and guidelines in the NIH Guide for the Care and Use of Laboratory Animals, the Animal Welfare Act, United States Department of Agriculture and the United States Public Health Service Policy on Humane Care and Use of Laboratory Animals. IACUC approved work with infectious Herpes Simplex Virus (HSV) strains under BSL2 conditions.

### Cells and viruses

The HSV-1 strains used in this study were HSV-1 strain 17syn+ [84] and HSV-2 strain 333. Immunization studies were carried out using HSV-2 *dl*5-29, which lacks the UL5 and UL29 genes and is derived from HSV-2 186 syn+ [65]. "Mockulum" mock infections were done using equivalent dilutions of uninfected Vero cell lysate.

Viruses were titered by standard plaque assay. Virus stock preparation and the plaque assay were performed using Vero cells as described previously [85]. Vero cells were cultured in Dulbecco's modified Eagle's medium supplemented with 5% fetal bovine serum, 250 U/ml penicillin, and 250 µg/ml streptomycin.

### Mice and animal procedures

All procedures were performed in accordance with federal and university policies. The mice used in this study were C57BL/6 (B6) and 5xFAD (MMRRC stock #: 34848) mice. The B6 mice were purchased from The Jackson Laboratory (Bar Harbor, ME) and the 5xFAD mice were provided by Charles Sentman, (Dartmouth), and bred in the barrier facility in the Center for Comparative Medicine and Research at the Geisel School of Medicine at Dartmouth.

### Viral challenge and neonatal monitoring

Neonatal mice postnatal day 1 (P1) were infected intranasally (i.n.) with $10^2$–$10^3$ PFU HSV or Vero cell lysate "mockulum" in a volume of 5 µl under 1% isoflurane anesthesia. Pups were monitored for weight loss or survival. Endpoints for survival studies were defined as excessive morbidity (hunching, spasms, or paralysis) and/or >10% weight loss (Fig 1B). Pups that were weighed daily were not used in the survival analysis as daily weighing can affect survival during the neonatal period. For behavioral studies, 100 PFU strain 17 (st17) was used as a control nHSV-1 infection. 75 PFU HSV-2 strain 333 was used as a control for nHSV-2 infection.

### Immunization

For the *dl*5-29 immunization, 6-week-old B6 female mice were immunized twice intramuscularly (i.m.), as done previously [64], with $10^5$ PFU of *dl*5-29 virus (see *Cells and Viruses*) in a 25 µl volume. Injections were carried out 21 days apart and with mice under 1% isoflurane anesthesia.

## Behavioral tests

Breeding and infections were conducted in a separate space than the behavioral studies room. Mice were acclimated to the behavioral facility under a 12-hour light cycle (7a-7p) for at least 7 days prior to testing. Environmental conditions (test room lighting, temperature, and noise levels) were kept consistent and both male and female mice were included in the study. Singly-housed animals were excluded from behavioral assessment as isolation is known to alter behavioral task performance [86]. Different mice were used for each behavioral assay to avoid confounding effects of training and participation on one behavioral task on performance in other assays.

## Novel object recognition

For the novel object recognition (NOR) test, animals were placed in a 30 cm x 30 cm open field containing two identical objects for 10 minutes [87]. Twenty-four hours later, mice were returned to the enclosure with one familiar object (from the previous day) and one novel object. Mouse movement in the enclosure was recorded (Canon VIXIA HF R800), and the amount of time mice spent with each object was quantified using open-source Matlab software that had been previously validated [88], or Noldus Ethovision [89]. Discrimination index ($Time_{novel} - Time_{familiar}$)/$Time_{total}$) was quantified for the 5-minute test trial to compare novelty preference while controlling for differences in individual mice's locomotion. The NOR test was conducted at 4–5 months post-infection. All enclosures and objects were cleaned thoroughly with 70% ethanol between sessions to eliminate olfactory cues. Inherent object preference was tested over a span of 5 minutes using naïve mice and object pairs were used only if no object preference was observed. Objects used in this study included: white plastic unicorn ducks, T-25 flasks filled with colorful aquarium pebbles, white plastic ducks, and pink plastic pig toys. Objects were balanced based on size and randomized for the familiar object training.

## Novel object location recognition

For the novel object location (NOL) task [46], mice were placed in the same 30 cm x 30 cm enclosure for 10 minutes during the familiarization phase. As illustrated in Fig 2, mice were familiarized to two identical objects in two corners of the arena (example: horizontal in the top two corners of the arena). One hour later, mice were returned to the enclosure with the same identical objects, one moved to a novel location in the arena (example: diagonal with one object remaining in the top left and one object moved to the bottom right). Mouse interaction time was recorded for 5-minutes (Canon VIXIA HF R800) and the first 20 seconds of total object interaction time was used to compute the discrimination index for each mouse (see above). All analysis was done using Noldus Ethovision software [89]. All enclosures and objects were cleaned thoroughly with 70% ethanol between sessions to eliminate olfactory cues. Object locations were balanced between familiarization and testing phases. Objects used in this study: saltshakers filled with DRIERITE desiccant.

## Modified Barnes maze

The modified Barnes maze (MBM) was performed on 5-month-old adult mice as previously described [57]. As illustrated in Fig 3, the maze consisted of a white 88.9 cm-diameter circle with 12 equally spaced holes around the circumference of a 15.24 cm-tall circular wall. Spatial cues consisting of large red letters ("W", "S", "T", "O") were positioned at 90-degree increments around the inside of maze. Mice were placed in the center of the platform facing the W and given 4 minutes to find the escape hole. If the mouse entered the escape hole before 4 minutes, the experiment ended. Mice that did not enter the escape hole were led to it by

the experimenter and allowed to exit. Mice received a total of 3 trials per day with an inter-trial interval (ITI) of 1 hour. Training consisted of 4 consecutive days when mice learned to find and enter the training escape hole. Directly following the 4 total days of training, the escape hole was switched to the location directly opposite its training position and mice were assessed for 2 days of 3 trials each. Mouse movement in the training and reversal trials was recorded (Canon VIXIA HF R800) and analyzed using Noldus Ethovision [89]. Escape latency and distance traveled were quantified in a blinded fashion using Noldus Ethovision software, while total errors and errors per quadrant (nose-pokes into non-escape holes) were quantified manually by two blinded experimenters.

## Different paired associate learning (dPAL)

Different paired-associate learning (dPAL) was conducted using the Second Generation Bussey-Saksida Touch Screen Chambers for Mice as previously described [60,62,90] (Lafeyette Instrument, model 80614A). Briefly, the testing was conducted in a trapezoidal operant chamber with a metal floor, a reward delivery machine, a touch screen, two infrared (IR) beams to detect mouse movement, and black Perspex sidewalls (see Fig 4A). The environment and data collection were controlled by ABET software provided by Campden Instruments. For the current experiment, a three-window (each window 7 cm x 7 cm) mask was placed over the touch screen to limit interaction zones.

To motivate mice to respond to a food reward, mice were food restricted and maintained at 85–90% initial free-feeding body weight. Weights were taken daily for one week prior to food restriction and every day of the experiment to ensure mice maintained appropriate body weight. Mice were rewarded with evaporated milk (Nestle Carnation) during each training and testing session. Mice were trained to interact with the touchscreen through four serial training stages: "Initial Touch" (criterion: completion of 30 trials within 60 minutes), "Must Touch" (criterion: completion of 30 trials within 60 minutes), "Must Initiate" (criterion: completion of 30 trials within 60 minutes), and "Punish Incorrect" (criterion: 80% correct responses or better within 60 minutes for two consecutive days). Once mice had successfully completed all training stages (average time, approximately two weeks), they graduated to dPAL testing.

The following dPAL protocol for mice was based on the work of John Talpos et al. [62], using the "Flower-Plane-Spider" stimulus combination shown in Fig 4. In dPAL, a stimulus, such as Flower, appeared in its correct touchscreen location (S+). A second stimuli, such as Plane, appeared in the incorrect location (S−). Response to S+ triggered a reward tone, illuminated the reward chamber, delivered milk reward, and removed visual stimuli from the screen. Response to S− triggered a 10 s time-out when stimuli was removed from the screen, the house-light was illuminated for 5 s, and an incorrect tone played. Following an S− response and after a 10 second ITI, the reward tray light illuminated and mice nose-poked to initiate a correction trial. Mice repeated correction trials until they picked S+ and proceeded to the next dPAL trial. The session finished after either 36 trials were completed, or 60 minutes had passed. Mice completed one testing session per day. Data was binned into 3-session blocks representing 108 trials each.

## Immunofluorescence

Brains were collected from PBS perfused mice, fixed with 4% paraformaldehyde, and cryopreserved in a sucrose gradient of 15%−30% at 4°C until they equilibrated and sank. Brains were then embedded in Tissue-Tek OCT compound (Sakura) and stored at −80°C prior to cutting. Slices were prepared using a Leica CM1860 UV cryostat at −15°C to −20°C. Tissue was mounted onto Colorfrost Plus microscope slides (Fisher Scientific) and incubated with 0.01%

BSA and 0.25% Triton X-100 for 20 minutes, then 5% normal goat serum and 0.25% Triton X-100 for 30 minutes for permeabilization and blocking. Primary and secondary antibodies were suspended in 1% NGS and 0.1% Triton X-100. Antibodies used in this study include rabbit anti-HSV gC (DAKO), Alexa Fluor 647 Dk anti-rabbit (Jackson ImmunoResearch), and 4',6-diamidino-2-phenylindole (DAPI; Thermo Fischer Scientific; 1:5,000). Microscopy images were acquired on a Zeiss Axio Observer.Z1 microscope and images shown are representative of at least three experiments.

To quantify fluorescence, images of DAPI-staining were used to manually register one sagittal section per brain to a sagittal mouse atlas, creating gross anatomical region outlines for the isocortex, olfactory bulb, diencephalon, midbrain, caudal brain stem, and cerebellum using ImageJ. Viral staining intensity was then quantified within each region using ImageJ and normalized to the maximum intensity within each brain region. Normalized average staining intensity was plotted for the 1000 PFU HSV-1-infected mice.

## Viral DNA extraction and HSV-1 genome copy number quantification

To quantify HSV-1 viral DNA (vDNA) in the brain of infected mice, we assayed the midbrain (MB), olfactory bulbs (OB), hindbrain, and cortex of pups infected with 100, 500, and 1000 PFU at 5 dpi. We also assayed the trigeminal ganglia. Matched samples from mock-infected pups were used as a negative control. Genomic DNA from each individual sample was isolated via proteinase K digestion, phenol-chloroform extraction, and ethanol precipitation. Following successful isolation of whole genome DNA, the copy number of HSV-1 per TG or brain region was determined using primers for VP16 (UL48; forward, 5'-AACCACATTCGCGAGCACCT TAAC-3'; Reverse, 5'-CAACTTCGCCCGAATCAACACC AT-3'). qPCR reactions were performed using the following components to a final reaction volume of 20 μl: 1X Luna Universal qPCR Master Mix (New England Biolabs), forward and reverse primers at a final concentration of 0.25 μM, 100 ng of DNA, and 5% acetamide. The qPCR reaction settings were as follows: 1 cycle at 95°C for 60 s, 42 cycles at 95°C for 15 s and 60°C for 30 s, followed by a melt curve with a range of 60–95°C. To determine HSV copy number, a standard curve was generated by diluting purified HSV-1 st17 genome into background mouse whole genome DNA isolate in 10-fold dilutions ranging from $10^6$–$10^0$ copies. These dilutions were used to determine the total genome copy number in the isolated brain and TG samples. To normalize the quantification of VP16, qPCR of a single-copy mouse gene, adipsin, was performed. The mouse adipsin primer set used was: 5'-AGTGTGCGGGGATGCAGT-3' (forward), and 5'- ACGCGAGAGCCCCAGGTA-3' (reverse). A standard curve for adipsin was also generated by making standards of uninfected mouse DNA isolate that range from $10^6$–$10^1$ copies. Based on the qPCR amplification of both VP16 and adipsin in test samples, HSV-1 copy number was calculated as described previously [91]. All genome copy number values that fell below the limit of detection of our VP16 standard curve were set to a normalized value of 0.001 for graphing.

## RNA extraction and quantitative RT-PCR for assessment of HSV-1 LAT

RNA was extracted from proteinase K digested brain and TG samples using the RNeasy Lipid kit (Qiagen) as described by the manufacturer. Following RNA isolation, LAT transcript was quantified using the Luna Universal One-Step RT-qPCR Kit (New England Biolabs) and primers for the LAT exon region (Forward: 5'-CGGCGACATCCTCCCCCTAAG-3', Reverse: 5'-GACAGACGAACGAAACATTCCG-3'). Once LAT transcript was quantified, copy number was calculated as described above (see *Viral DNA extraction and HSV-1 genome copy quantification*). All LAT exon quantification values that fell below the limit of detection (LOD) of our LAT standard curve were set to a normalized value of 0.001 for graphing.

## Statistical analysis

Prism 10 (GraphPad) software was used for statistical tests. For survival studies, nHSV-infected mice were compared to mock-infected controls using the log rank Mantel-Cox test to determine p-values. For copy number analysis, HSV dosage groups were compared to each other via a one-way ANOVA with Bonferoni's test for multiple comparisons. For NOR, NOL, and MBM performance, groups were compared using one-way ANOVA with Tukey's multiple comparisons tests, one-way ANOVA with Dunnett's multiple comparisons tests, and unpaired t-tests. For weight gain, dPAL, and MBM assessments, groups and time points were compared via two-way ANOVA, with repeated measures design and Sidek's multiple comparisons test.

## Supporting information

**S1 Fig. Viral genome in the trigeminal ganglia and brains of 100 PFU nHSV-1 infected mice.** HSV genome copy number in the trigeminal ganglia (A), olfactory bulbs (B), cerebral cortex (C), and hindbrain (D) of 100 PFU nHSV-1 infected mice 5 dpi, relative to mouse single-copy adipsin gene, as determined by RT-PCR. (E) HSV genome copy number in the trigeminal ganglia, olfactory bulbs (OB), hindbrain, and cerebral cortex of 100 PFU nHSV-1-infected mice at 6 mpi. (F) Quantification of HSV-1 latency-associated transcript (LAT) in the trigeminal ganglia, olfactory bulbs (OB), hindbrain, and cortex of 100 PFU nHSV-infected mice at 6 mpi. Error bars represent standard deviation from the geometric mean.
(TIF)

**S2 Fig. Behavioral morbidity following nHSV-2 infection.** (A) Performance of mock-infected (n = 18), strain 17 (st17) nHSV-1-infected (n = 20), and strain 333 nHSV-2-infected mice (n = 12) on the novel object recognition (NOR) task at 5 mpi. Bar graph shows the mean discrimination index for the novel object. Discrimination index = (Time$_{novel}$ − Time$_{familiar}$)/Time$_{total}$. (B-D) Performance of mock-infected (n = 10), strain 17 (st17) nHSV-1-infected (n = 5), and strain 333 nHSV-2-infected mice (n = 4) on the MBM task at 5 mpi. Mean latency to escape on the first trial of reversal testing in the modified barnes maze (MBM). (E) Mean distance traveled in centimeters (cm) during the first trial of reversal testing in the MBM. (F) Mean errors per trial committed during the first trial of reversal testing in the MBM. Statistical significance determined by one-way ANOVA with Tukey's and Dunnett's multiple comparisons post-hoc tests (D-E). *p < 0.05, **p < 0.01. Error bars represent standard error (SEM).
(TIF)

**S3 Fig. Viral genome in trigeminal ganglia and brains of 100 PFU nHSV-1 infected offspring of *dl*5-29-vaccinated dams.** HSV genome copy number in the trigeminal ganglia (A), olfactory bulbs (B), cerebral cortex (C), and hindbrain (D) of 100 PFU nHSV-1 infected pups of *dl*5-29-vaccinated dams 5 dpi, relative to mouse single-copy adipsin gene, as determined by RT-PCR. Statistical significance was determined by unpaired t-test. ns = not significant. Error bars represent standard deviation from the geometric mean.
(TIF)

## Acknowledgments

We thank all members of the Leib and Nautiyal Labs, Margaret Ackerman, Pamela Rosato, Alex Skorput, Kirk Maurer, Bryan Luikart, Steve Fiering, Charles Sentman, Hermes Yeh, the late David Bucci, and the Jones Media Center for materials and/or helpful discussion.

## Author contributions

**Conceptualization:** Abigail J Dutton, Evelyn M Turnbaugh, Chaya D Patel, David M Knipe, Katherine M Nautiyal, David A. Leib.

**Data curation:** Abigail J Dutton, David A. Leib.

**Formal analysis:** Abigail J Dutton, Evelyn M Turnbaugh, Chaya D Patel, Sean A Taylor, Katherine M Nautiyal, David A. Leib.

**Funding acquisition:** David M Knipe, David A. Leib.

**Investigation:** Abigail J Dutton, Evelyn M Turnbaugh, Chaya D Patel, Callaghan R Garland, Sean A Taylor, Roberto Alers-Velazquez.

**Methodology:** Abigail J Dutton, Evelyn M Turnbaugh, Chaya D Patel, Callaghan R Garland, Sean A Taylor, Roberto Alers-Velazquez, Katherine M Nautiyal.

**Project administration:** David A. Leib.

**Resources:** David M Knipe.

**Software:** Katherine M Nautiyal.

**Supervision:** Katherine M Nautiyal, David A. Leib.

**Validation:** Abigail J Dutton, Evelyn M Turnbaugh, Callaghan R Garland, Sean A Taylor.

**Writing – original draft:** Abigail J Dutton, David A. Leib.

**Writing – review & editing:** Abigail J Dutton, Evelyn M Turnbaugh, Chaya D Patel, Callaghan R Garland, Sean A Taylor, Roberto Alers-Velazquez, David M Knipe, David A. Leib.

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
