## [Decision Letter · Decision Letter 0]

13 Nov 2024

PPATHOGENS-D-24-02176Asymptomatic neonatal herpes simplex virus infection in mice leads to long-term cognitive impairmentPLOS Pathogens Dear Dr. Leib, Thank you for submitting your manuscript to PLOS Pathogens. After careful consideration, we feel that it has merit but does not fully meet PLOS Pathogens's publication criteria as it currently stands. Therefore, we invite you to submit a revised version of the manuscript that addresses the points raised during the review process. All three reviewers thought this manuscript was highly important, significant and key in establishing a paradigm for HSV-1 related neurodegeneration. Each of the reviewers have suggested key experiments that will substantially improve the impact of the paper. These include (but are not limited to) inclusion of additional controls to decisively conclude that virus is undetectable in brains (line 349), assessment of the infection status of adult mice subjected to cognitive tests and evidence of histological or inflammatory changes observed following primary infection. In addition, please be sure to address each of the additional comments made by the reviewers. Please submit your revised manuscript within 60 days. If you will need more time than this to complete your revisions, please reply to this message or contact the journal office at plospathogens@plos.org. Please include the following items when submitting your revised manuscript:* A rebuttal letter that responds to each point raised by the editor and reviewer(s). You should upload this letter as a separate file labeled 'Response to Reviewers '. This file does not need to include responses to any formatting updates and technical items listed in the 'Journal Requirements' section below.* A marked-up copy of your manuscript that highlights changes made to the original version. You should upload this as a separate file labeled 'Revised Manuscript with Track Changes '.* An unmarked version of your revised paper without tracked changes. You should upload this as a separate file labeled 'Manuscript '. If you would like to make changes to your financial disclosure, competing interests statement, or data availability statement, please make these updates within the submission form at the time of resubmission. Guidelines for resubmitting your figure files are available below the reviewer comments at the end of this letter. We look forward to receiving your revised manuscript. Kind regards, Donna M NeumannAcademic EditorPLOS Pathogens Blossom DamaniaSection EditorPLOS Pathogens Michael Malim

Editor-in-Chief

PLOS Pathogens

orcid.org/0000-0002-7699-2064 **Additional Editor Comments (if provided):** All three reviewers thought this manuscript was highly important, significant and key in establishing a paradigm for HSV-1 related neurodegeneration. Each of the reviewers have suggested key experiments that will substantially improve the impact of the paper. These include (but are not limited to) inclusion of additional controls to decisively conclude that virus is undetectable in brains (line 349), assessment of the infection status of adult mice subjected to cognitive tests and evidence of histological or inflammatory changes observed following primary infection. In addition, please be sure to address each of the additional comments made by the reviewers.  **Journal Requirements:****Reviewers' Comments:** Reviewer's Responses to Questions

**Part I - Summary**

Reviewer #1: The study investigated whether cognitive impairment could result from asymptomatic herpes simplex virus infections. To do so, the authors employed a low PFU model of HSV-1 infection, one that results in low-grade morbidity (90% survival) with detection of HSV-1 genomes but not antigen (gC) in the brain. Cognitive impairment was assessed using the novel object recognition (NOR) task, the novel object location (NOL) task, the modified Barnes maze (MBM), and the different paired-associate learning (dPAL) task. The authors tested the effects of viral strain (HSV-1 vs. HSV-2) and maternal immunization in the context of their neonatal infection model. The authors found that HSV infection resulted in decreased performance in NOR, NOL, MBM, and dPAL tests. The difference within NOR tasks was comparable to a genetic mouse model of Alzheimer’s disease (Fig. 2A). Within the context of the MBM assessment, differences between mock and HSV-1 infected mice was only detected at day 1 of the reversal, with no major differences observed in training or day 2-6 of reversal (Fig. 3). Finally, maternal immunization prior to HSV-1 infection resulted in protection from cognitive impairment as assessed via the NOR task. The manuscript is well written with conclusions that logically follow the data. The potential contribution of HSV1 to neurodegeneration is a key question within the field and we agree that this low PFU model tackles the question in a new way. This manuscript is an important starting point from which additional viral/host factors and neuro/immunologic characterization can be performed.

Reviewer #2: Several lines of evidence suggest that herpes simplex virus (HSV) infection of the central nervous system (CNS) and subsequent inflammation is associated with neurodegeneration and cognitive impairment, contributing to dementia, including Alzheimer’s disease (AD). For instance, post-mortem analyses revealed higher level of HSV DNA in the CNS of AD patients than in non-AD patients. The worldwide prevalence of HSV infection in the adult population is high, about 70% for HSV-1 and approx. 13% for HSV-2. The authors hypothesize that since sub-clinical HSV reactivations are frequent, there is the possibility of undetected asymptomatic neonatal infections that have long-term consequences in the nervous system, in particular neurological and cognitive damage. This could explain, at least partially, the potential relationship between HSV infection and dementia. To address their hypothesis, the authors infect neonatal mice (day 1 post-birth) intranasally with several HSV doses to determine one that would lead to subclinical infection of the central nervous system. Then, they infect neonatal mice intranasally with this dose (100 plaque forming units (p.f.u.)) and perform several behavioural and memory test with mock- and HSV-1- as well as HSV-2-infected mice. The results clearly show that sub-clinical HSV infection results in cognitive impairment. Interestingly, despite HSV-2 normally causing more severe neonatal infections than HSV-1, the authors did not observe any different between the two viruses when assessing long-term neurological complications. They also observed similar impairment of cognitive function when comparing infection with 5xFAD mice, commonly employed as a model to study AD. Finally, the authors showed that vaccination of pregnant dams with replication-deficient HSV-2 vaccine (strain dl5-29) inhibited HSV infection of the CNS and protected from behavioral impairment.

Overall, this report addresses a highly relevant topic, the potential implication of HSV infection in the development of cognitive impairment. The authors employ mouse models to mimic neonatal infection in humans and perform several behavioral and cognitive tests to determine the relevance of subclinical neonatal infection on neurological function. They also show that vaccination of the mothers could protect from CNS infection and disease. The manuscript is very well written and explained. The rational for the experiments is clear and the results support most of the conclusions. I only have a few comments that I think could improve this manuscript.

Reviewer #3: HSV is a highly prevalent pathogen of the peripheral nervous system, but HSV can be neuroinvasive and has been implicated as a potential contributor to Alzheimer’s disease. HSV infection in neonates can cause severe disease. The rate at which neonatal HSV infection occurs but is asymptomatic is harder to define but given the high prevalence of HSV it seems likely that subclinical neonatal HSV infections are not uncommon. Here, Dutton and colleagues infected newborn mice with a low dose of HSV, such that the mice survive the infection, then when the mice are adults assessed cognitive function and behavior, and find long-term deficits in mice infected as neonates. They show that these deficits require infection, as they are prevented by maternal vaccination and not induced by inoculation of neonates with a vaccine strain. This work builds on prior studies from this group that investigated outcomes in younger mice infected as neonates (the present work also uses a more extensive set of cognitive and behavioral tests) and provides important insights into how viral infections can have durable impacts on cognitive and behavioral function. Overall the manuscript is well-written and nicely presented.

**Part II – Major Issues: Key Experiments Required for Acceptance**

Reviewer #1: 1. Immunofluorescence (Fig. 1C, 5B):

a. The authors only show one representative images for their IF data (Fig. 1C, 5B). Please perform image analysis to quantify the gC fluorescence per image in each condition. If possible the gC+ would be further quantified per brain region, e.g. brainstem, midbrain, cortex.

b. As a companion to Fig. 1C, include both an image and quantification (see comment #3) for the lack of gC detection in 100 PFU infected brains. This is to support the text claim at line 193-194.

c. The authors stained for gC which would only be detected at late stages during productive infection. This does not rule out abortive lytic infection vs. latency. In an ideal world the authors would also stain for IE or E viral proteins (e.g. ICP4, ICP8). However in lieu of that, we would ask the authors to clarify in the text what the presence of gC means in the context of infection and give those respective caveats.

2. Characterization of HSV1 CNS infection after maternal immunization.

a. The claim at line 349-352 is overstating the data presented “Given that virus was undetectable in brains of offspring from vaccinated dams at 5 dpi, the development of neurologic morbidity is likely dependent on HSV establishing a threshold of acute and latent neonatal CNS infection.”

b. You cannot state “virus was undetectable in brains” (line 349-350) when you only show data for HSV1 DNA in the midbrain. Please include data for other regions similar to what you show in Sup. Fig. 1 (TG, olfactory bulb, cortex, hindbrain)

c. With the current data presented, it’s unclear whether the virus fails to establish latency after maternal immunization or if it does not reach the CNS during primary infection. Please comment.

3. Low PFU model=asymptomatic

a. The only data to demonstrate this is the survival curve in Fig. 1B, which still has 10% of the animals succumbing. We would ask the authors to include additional metrics for the mice to support the model as asymptomatic.

Reviewer #2: 1. The authors suggest that primary infection could cause inflammation or affect proper neurological development leading to long-term morbidity. They authors should perform assays to determine whether this is the case (e.g., histology, quantification of proteins involved in inflammation, expression of appropriate neuronal and glial cell markers, neuronal death, etc). In case such changes are not observed, how do the authors explain the long-term consequences of infection?

2. Are there any anatomical and/or physiological changes associated with dementia in the CNS of mice with cognitive impairment?

Reviewer #3: The authors assess virologic outcomes in the infected neonates (viral loads by qPCR, viral antigen staining in brains), but there is no assessment of the infection status of the adult mice subjected to cognitive tests. In this model, do the mice clear the infection or do they have a latent (or persistent) infection? Tissues such as brain and trigeminal ganglia should be collected from adult mice infected as neonates and viral genomes and transcripts measured by qPCR and RT-qPCR (these tissues potentially could be harvested after cognitive testing is completed to allow correlations between virologic outcomes and cognitive performance).

**Part III – Minor Issues: Editorial and Data Presentation Modifications**

Reviewer #1: 1. Please remove the text at lines 387-389 “The small dose of HSV inoculum needed to induce memory impairment suggests that viral-induced cognitive decline may not depend on reactivation events in our model.” While the model did not include or test any intentional reactivation triggers, the authors cannot assume spontaneous reactivation isn’t occurring in their model & contributing to the cognitive changes detected. The authors did not look for HSV-1 in the brain at any time points other than 5 dpi, nor did they test for viral shedding in the period between infection and cognitive testing.

2. The authors fail to comment on why a difference between mock and HSV-1 infection were only seen at day 1 via the MBM test. What would this result indicate regarding the actual cognitive delay or type of damage? Please include a discussion of this.

3. Data presentation:

a. All bar graphs (Fig. 1D, Fig. 2 bottom panels, Fig. 3 D-G, Fig. 4D-F, Fig. 5C-E, Sup Fig. 1A-D, Sup Fig. 2 A-D) need to be updated to show individual data points for each mouse/sample included.

b. Include “ns” labels for statistical tests performed that are not significant (Fig. 3G, 4B-C, Sup Fig. 2B-C)

Reviewer #2: 1. Is the text referring to the statistics employed in Figure 1A correct? Or does it refer to panel B?

2. It will be more informative to show the performance of each individual mouse in the tests with a symbol, rather than using bars to represent the mean/average.

3. It is not clear if the same mice undergo each individual test or whether different mice were employed in the different tests. Please, clarify this. If the same mice were used in different tests, do the same individual mouse always perform worse/better in the test? Or are there mice that perform well in some tests and bad in others?

4. Was viral DNA detected in tissues/organs other than CNS in the offsprings of vaccinated dams following challenge with HSV?

5. Line 418: Typo in “Additinally”

6. Line 419: “antibody prevents” should be “antibodies prevent”.

7. Line 483: Dr. David Knipe is a co-author of this manuscript so I suppose it is not necessary to indicate that he provided the replication-deficient virus. Similarly, he should not be included in the Acknowledgements.

8. Line 602 and others: Separate the cipher from the unit of measure (i.e., 10s --- 10 s).

9. Supplemental Figure 1: Include trigeminal ganglion in the title of the figure, this is not a component of the brain.

Reviewer #3: The authors justify their use of the low-dose neonatal infection model by noting that asymptomatic shedding is common for genital herpes. But in this case, it is the infected mother who is asymptomatic, not necessarily the neonate, who may simply be uninfected since asymptomatic shedding likely occurs in the context of recurrent maternal infection, rather than a primary infection, in which case maternal antibodies likely provide substantial protection against neonatal infection. Altogether, it is not necessary to use asymptomatic shedding as a justification for this mouse model or study (the model is fine and it is reasonable to investigate outcomes of subclinical neonatal HSV infections).

In addition to noting prior work assessing cognitive and behavioral outcomes in mice after HSV infection, the authors may consider discussing studies with other viruses, particularly with neonatal and congenital infection models. Examples of references to consider: McMahon 2024 JCI Insight 38781563; Figueiredo 2019 Nat Commun PMID 31488835; Souza 2018 Sci Transl Med PMID 29875203; Vasek 2016 Nature PMID 27337340; Garber 2019 Nat Neurosci PMID 31235930.

Throughout: change “symptoms” to “clinical signs”, “disease signs”, or similar when referring to mice.

Throughout: replace bar charts with scatter plots to better convey data distribution

Consider moving HSV-2 data into the main figure with HSV-1 (supplementary Fig 2).

Line 169-170: consider deleting “in this era of vaccine hesitancy” (not necessary)

Line 197: trigeminal ganglia are not part of the brain

Line 423-424: delete this sentence (not necessary)

Line 427-430: Consider tempering this claim. The present study concludes that viral infection (and presumably replication in the CNS) is necessary to induce impairments so the concern raised here may not be applicable to some vectors.

Line 483, 671: it is not necessary to thank Dr. Knipe as he is an author on the study.

Line 495: is this the correct JAX# for these mice?

PLOS authors have the option to publish the peer review history of their article (what does this mean? ). If published, this will include your full peer review and any attached files.

**Do you want your identity to be public for this peer review?** For information about this choice, including consent withdrawal, please see our Privacy Policy .

Reviewer #1: No

Reviewer #2: No

Reviewer #3: No

---

## [Editor Report · Decision Letter 1]

25 Jan 2025

Dear Dr. Leib,

We are pleased to inform you that your manuscript 'Asymptomatic neonatal herpes simplex virus infection in mice leads to persistent CNS infection and long-term cognitive impairment' has been provisionally accepted for publication in PLOS Pathogens.

Best regards,

Donna M Neumann

Academic Editor

PLOS Pathogens

Blossom Damania

Section Editor

PLOS Pathogens

Sumita Bhaduri-McIntosh

Editor-in-Chief

PLOS Pathogens

orcid.org/0000-0003-2946-9497

Michael Malim

Editor-in-Chief

PLOS Pathogens

orcid.org/0000-0002-7699-2064
---

## [Editor Report · Acceptance letter]

Dear Dr. Leib,

We are delighted to inform you that your manuscript, "Asymptomatic neonatal herpes simplex virus infection in mice leads to persistent CNS infection and long-term cognitive impairment," has been formally accepted for publication in PLOS Pathogens.

Best regards,

Sumita Bhaduri-McIntosh

Editor-in-Chief

PLOS Pathogens

orcid.org/0000-0003-2946-9497

Michael Malim

Editor-in-Chief

PLOS Pathogens

orcid.org/0000-0002-7699-2064